# Insertional Mutagenesis as a Strategy to Open New Paths in Microalgal Molybdenum and Nitrate Homeostasis

**DOI:** 10.3390/cimb47060396

**Published:** 2025-05-26

**Authors:** Esperanza Leon-Miranda, Manuel Tejada-Jimenez, Angel Llamas

**Affiliations:** Department of Biochemistry and Molecular Biology, Campus de Rabanales and Campus Internacional de Excelencia Agroalimentario (CeiA3), Edificio Severo Ochoa, University of Córdoba, 14071 Córdoba, Spain

**Keywords:** *Chlamydomonas*, nitrogen, transformation, ABC superfamily, BAT1, Moco

## Abstract

Molybdenum (Mo) is a vital micronutrient for nearly all living organisms, serving as a cofactor for molybdoenzymes that catalyze essential redox reactions in nitrogen metabolism. Among these enzymes, nitrate reductase plays a crucial role in nitrate assimilation. Maintaining Mo homeostasis—including uptake, storage, and utilization—is critical to avoid both deficiency and toxicity. Our research focuses on uncovering novel molecular components involved in Mo homeostasis, particularly in connection with nitrate assimilation, using *Chlamydomonas reinhardtii*, a model green microalga. To achieve this, we generated more than 5000 *Chlamydomonas* transformants through insertional mutagenesis using a paromomycin resistance cassette (*AphVIII*) and screened them for altered growth on nitrate and under different Mo concentrations. We identified four strains showing altered growth patterns when using nitrate as a nitrogen source or exhibiting increased sensitivity or resistance to Mo. The genomic alterations in these strains were identified. Notably, both a Mo-resistant and a Mo-sensitive transformant had disruptions in the genes that encoded ABC-type transport proteins, indicating a potential role for these proteins in Mo transport. Additionally, two strains were unable to grow on nitrate. One of them had a mutation in the *CNX7*, a gene involved in Mo cofactor biosynthesis, while the other had a mutation in *BAT1*, an amino acid transporter. The *BAT1* mutant represents an interesting case study, as this gene has not previously been associated with nitrate metabolism. These findings enhance our understanding of Mo and nitrate homeostasis mechanisms and open new paths for engineering microalgae with improved nitrogen assimilation.

## 1. Introduction

Microalgae have emerged as a significant source of innovation in various fields, including bioremediation, biofuel production, biomass generation, biofertilizer development, and the creation of high-value-added products [1]. Among microalgae, the unicellular green alga *Chlamydomonas reinhardtii* (hereafter referred to as *Chlamydomonas*) has become a key focus of study, with numerous important advancements being made across these areas [2]. *Chlamydomonas* is used as a model organism for studying various processes, including chloroplast dynamics, photosynthesis, mineral nutrition, cell cycles, and flagellar biology, among others [3]. Some advantages of using *Chlamydomonas* are that it is easy to cultivate under various conditions, including in synthetic mineral media, and it reproduces rapidly, with a generation time of about 8 h, allowing for quick biomass production [4]. Moreover, a wide variety of laboratory techniques are available for working with *Chlamydomonas*, including methods in molecular biology, physiology, and biochemistry. Additionally, *Chlamydomonas* has a fully sequenced haploid genome, and stable mutants can be generated and isolated [5].

Nitrogen (N) acquisition is a fundamental process for all living organisms, including crop plants, where N is often a limiting factor that determines productivity. In soils, nitrate is typically the preferred form of inorganic N available to plants, which is why it is commonly used in fertilizers. Once inside the plant, nitrate is reduced to nitrite by the enzyme nitrate reductase (NR), which is then further reduced to ammonium and incorporated into carbon skeletons to form amino acids [6]. NR functions as a homodimer, and each monomer contains, as prosthetic group, molybdenum cofactor (Moco) [7]. The importance of molybdenum (Mo) in living organisms was first studied by Arnon and Stout in 1949 using tomato plants [8]. Subsequent research has established the role of Mo in promoting plant growth and its involvement in redox processes related to N metabolism [9]. All cells take up Mo in the form of the oxyanion molybdate (MoO_4_^2−^), which is the most abundant form of Mo in aqueous solution at pH above 4.2 [10]. Mo plays a crucial role in the active sites of various enzymes (molybdoenzymes) found in bacteria, fungi, plants, and animals [11]. In bacteria, Mo uptake is mediated by different protein systems from the ABC superfamily [12]. In eukaryotes, two molybdate transport systems are known: MOT-1 and MOT-2, both first identified in *Chlamydomonas*. Neither of these proteins shows similarity to members of the ABC superfamily [13,14]. In general, molybdoenzymes catalyze redox reactions that involve the transfer of two electrons to or from a substrate [15]. In eukaryotes, in addition to NR, four other molybdoenzymes have been identified: sulfite oxidase (SO), aldehyde oxidase (AO), xanthine dehydrogenase (XDH), and the mitochondrial amidoxime-reducing component (mARC) [16]. The mARC protein is involved in the reduction of N-hydroxylated compounds and has been reported to participate in the synthesis of nitric oxide [17]. SO detoxifies sulfite in sulfur metabolism [18]. In humans, Mo deficiency is primarily linked to the loss of SO activity, resulting in sulfite accumulation and disruptions in the catabolism of sulfur-containing amino acids. This leads to severe neurological degeneration, which typically causes premature death within the first few weeks of life [19]. XDH catalyzes purine degradation by oxidizing hypoxanthine to uric acid [20]. AO exhibits greater substrate specificity than XDH, with its substrates being purines, pteridines, aldehydes, and aromatic and aliphatic heterocycles [21]. In these enzymes, Mo must be coordinated with molybdopterin (MPT), which is a complex heterocycle featuring a pyran fused to a pterin ring, to become biologically active, forming the Moco [22].

All organisms require mechanisms to maintain an adequate concentration of essential metals within their cells [23]. Situations may arise where certain metals are either scarce, leading to deficiency, or present in excess concentrations, causing potentially lethal toxicity. Therefore, it is crucial to ensure the cellular availability of the metals while preventing the potential toxic effects of their accumulation [24]. This set of mechanisms is known as metal homeostasis, a highly regulated process that ensures proper metal uptake, storage, transport, and distribution [25]. It involves specialized mechanisms for absorption from the environment, intracellular compartmentalization, and long-distance transport [26]. For this reason, organisms have developed metal storage systems through compartmentalization or by binding to small molecules, such as proteins or organic acids, which help to create metal reserves while preventing their toxic effects [27].

In plants, a deficiency of Mo, and, consequently, its cofactor, leads to an inability to grow on nitrate as a N source [28]. Engineering plants for efficient N assimilation is a major challenge, but it has the potential to significantly reduce the need for intensive N fertilization in agricultural soils [29]. To achieve this goal, it is essential to understand the various elements involved in Mo homeostasis, as it has a critical role in nitrate assimilation. However, most of the Mo homeostasis components are still unknown in plants. Therefore, our primary objective is to identify new potential molecular components involved in Mo and nitrate homeostasis using *Chlamydomonas* as a model organism through insertional mutagenesis. Mutants affected in genes involved in Mo homeostasis are likely less efficient in dealing with Mo toxicity or deficiency conditions. To obtain Mo- and nitrate-related mutants, a DNA cassette conferring resistance to the antibiotic paromomycin (*AphVIII*) was randomly inserted into the *Chlamydomonas* genome. The growth ability of the resulting paromomycin-resistant transformants was evaluated under different Mo and N conditions compared to the parental strain. Four transformants were selected for further analysis: two exhibited impaired growth on nitrate and two displayed abnormal responses to Mo, showing either enhanced resistance or increased sensitivity. The identification of the affected genes in these mutants revealed two ABC-type proteins potentially involved in Mo transport. Additionally, two genes related to nitrate metabolism were identified: *CNX7*, which is involved in Moco biosynthesis, and *BAT1*, a putative amino acid transporter whose role in nitrate metabolism remains to be elucidated.

## 2. Materials and Methods

### 2.1. Chlamydomonas Strains and Growth Conditions

The *Chlamydomonas* strains used in this study included the cell wall-deficient strain 704, which was obtained from the Chlamydomonas Resource Center (https://www.chlamycollection.org/, accessed on 20 May 2025) [30], and the transformants generated using the *AphVIII* marker. Cells were cultured for 3–6 days in a temperature-controlled chamber at 25 °C under continuous light (15–30 W/m^2^) provided by white LED panels. Cultures were grown in Tris-Acetate-Phosphate (TAP) medium following previously described methods [4]. Depending on the required volume, 25 mL, 50 mL, or 100 mL Erlenmeyer flasks were used as culture vessels. Strains were maintained on solid TAP medium in sterile Petri dishes and refreshed every 2–3 months to ensure viability. For Mo-free conditions, a trace element solution prepared without adding Mo was used.

### 2.2. Growth Test

To analyze growth differences in solid media, following transformation, agar plates were incubated in a growth chamber for 7 days. Colonies growing on plates containing both ammonium and paromomycin were selected with sterile toothpicks and resuspended in 20 μL of liquid culture medium. Cell concentration was determined, and 5 μL containing approximately 500 cells was spotted onto agar plates with the corresponding media. After 7 days of incubation, plates were photographed to assess colony development. To analyze growth differences in liquid media, a sequential process was carried out, including a pre-inoculum step. For the pre-inoculum, the different *Chlamydomonas* strains under study were taken from solid agar cultures and transferred to liquid medium. These liquid cultures were kept in the growth chamber for 2–3 days until they reached the exponential growth phase. Afterwards, the cells were centrifuged at 3000× *g* for 1 min, washed, and resuspended in fresh medium. Cell concentration was adjusted to 50,000 cells/mL for each condition. After 7 days of growth, chlorophyll concentration was analyzed as an indicator of biomass accumulation.

### 2.3. Quantification of Chlamydomonas Cell Growth

*Chlamydomonas* growth was quantified using two complementary approaches: direct cell counting and chlorophyll content measurement. To determine the cell number, an automatic cell counter (Countess 3 FL, Invitrogen, Waltham, MA, USA) was used. To quantify chlorophyll concentration, a 1 mL sample of cells was centrifuged, and the cell pellet was resuspended in 1 mL of 100% (*v*/*v*) acetone. The suspension was incubated at 4 °C in the dark for 8 h to allow complete pigment extraction. The sample was then centrifuged again, and the absorbance of the supernatant was measured at 652 nm to estimate total chlorophyll content [31].

### 2.4. Preparation of the Selectable Marker for Insertional Mutagenesis

For the generation of insertional mutants, the *Chlamydomonas* strain 704 was transformed using the *AphVIII* gene as a selectable marker. This gene confers resistance to the antibiotic paromomycin and has been used in other functional genomics studies in *Chlamydomonas* [32]. The *AphVIII* gene, derived from *Streptomyces rimosus*, encodes an aminoglycoside 3′-phosphotransferase type VIII enzyme that inactivates aminoglycoside antibiotics such as paromomycin. To allow this gene to be expressed in *Chlamydomonas*, its coding sequence was placed under the control of a chimeric *Chlamydomonas* promoter, consisting of the small subunit of rubisco (*RbcS2*) promoter and the *Hsp70A* terminator, which together provide robust transcriptional activity. The resulting plasmid was named pSI104 [33]. For the PCR amplification of the *AphVIII* gene from the pSI104 plasmid, the following primers and conditions were used: Primer pair Parcomp1-F (5′-GCGGGGAGCTCGCTGAGG-3′) and Parcomp1-R (5′-CGCTTCAAATACGCCCAG-3′). Amplification was performed using Phusion™ High-Fidelity DNA Polymerase (Thermo Scientific, Waltham, MA, USA). The PCR program consisted of initial denaturation at 98 °C for 30 s, followed by 30 cycles of denaturation at 98 °C for 10 s and combined annealing and extension at 72 °C for 1 min, concluding with a final extension at 72 °C for 5 min (Figure 1A). The resulting PCR product was analyzed on an agarose gel, verified to correspond to the expected size (~2 kb), sequenced to confirm its identity, and quantified. This DNA, hereafter referred to as the *AphVIII* marker, was used in the subsequent transformation experiment for insertional mutagenesis.

### 2.5. Chlamydomonas Transformation

For the generation of *Chlamydomonas* transformants, an electroporation method was used with some modifications (Figure 1B) [34]. *Chlamydomonas* cells were cultured in liquid medium until they reached the exponential growth phase. They were then harvested by centrifugation at 3000× *g* for 2 min and resuspended in TAP medium to a final density of 3 × 10^6^ cells/mL. The transformation was carried out by exposing 250 µL of the cell suspension to an electrical pulse of 600 mV for 20 ms in a 4 mm gap electroporation cuvette. The amount of DNA used in the process was 10 ng. The electric pulse was applied using a Gene Pulser Xcell system (Bio-Rad, Hercules, CA, USA). After the electroporation procedure, the cells were kept in the cuvette for 10 min to facilitate initial recovery. Subsequently, the cells were transferred to a tube containing 10 mL of TAP medium supplemented with 40 mM sucrose and incubated at 25 °C on an orbital shaker at 50 rpm under continuous light for 12–16 h. After the recovery period, the cells were centrifuged at 3000× *g* for 2 min. The resulting pellet was resuspended in 500 µL of TAP medium with 40 mM sucrose and plated onto selective medium containing 25 µg/mL paromomycin.

### 2.6. Genomic DNA Isolation from Chlamydomonas

Genomic DNA was isolated using a modified phenol-chloroform method [35]. *Chlamydomonas* cells corresponding to 15 mL of an exponentially growing culture were harvested by centrifugation at 3000× *g* for 2 min. The cell pellet was resuspended in 700 µL of lysis buffer (50 mM Tris-HCl, pH 8.0, 5 mM EDTA, 300 mM NaCl). SDS (sodium dodecyl sulfate) was then added to a final concentration of 2%. The suspension was gently vortexed and incubated for 10 min at 4 °C to facilitate cell lysis. Subsequently, nucleic acid extraction was performed by adding an equal volume of phenol/chloroform/isoamyl alcohol (25:24:1 *v*/*v*/*v*) pre-saturated with 50 mM Tris-HCl, pH 8.0. The mixture was vortexed vigorously for 1 min and then centrifuged at 13,000× *g* for 5 min at 4 °C (the initial centrifugation was carried out for 10 min at 4 °C). The aqueous (upper) phase was carefully collected, and additional extraction was repeated as necessary until a clean interface was obtained. To remove residual phenol, an equal volume of chloroform saturated with water was added to the aqueous phase. The mixture was then vortexed vigorously for 1 min and centrifuged at 13,000× *g* for 5 min at 4 °C. Genomic DNA was then precipitated by adding two volumes of cold 100% ethanol and centrifuged at 13,000× *g* for 20 min at 4 °C. The nucleic acid pellet obtained was then washed with 500 µL of cold 70% ethanol and allowed to dry at room temperature for 10 min. Finally, the genomic DNA pellet was resuspended in 25 µL of nuclease-free water. To remove RNA contamination, 1 µL of ribonuclease (100 µg/mL) was added to each sample.

### 2.7. DNA Quantification

DNA was quantified using a NanoDrop™ 2000/2000c spectrophotometer (Thermo Scientific, Waltham, MA, USA). Additionally, DNA content was semi-quantitatively estimated through agarose gel electrophoresis. Samples were run alongside λ phage DNA standards of known concentration, and relative quantities were estimated by comparing band intensities under UV illumination.

### 2.8. Purification of DNA Fragments from Agarose Gels

DNA fragments of interest were excised using a sterile scalpel and transferred to a 1.5 mL tube. Gel-purified DNA was obtained using the NucleoSpin^®^ Gel and PCR Clean-up kit (Macherey Nagel, Düren, Gernamy), following the manufacturer’s instructions.

### 2.9. Identification of the Genomic Region Adjacent to the AphVIII Marker Insertion

The strategy used to identify the *Chlamydomonas* genomic regions adjacent to the insertion of the *AphVIII* marker was an adaptation of the Inverse Polymerase Chain Reaction method [36], with the following modifications. Isolated *Chlamydomonas* genomic DNA was digested with the restriction enzyme *AvaI* (New England Biolabs, Ipswich, MA, USA). This enzyme cuts very frequently in the *Chlamydomonas* genome but only once in the *AphVIII* marker. Genomic digestion was carried out using 1–5 μg of DNA and 10–20 units of *AvaI* enzyme in a final volume of 20 μL, with incubation for 8–16 h at 37 °C. This digestion generated two DNA fragments per insertion site: one containing the 5′ region of the *AphVIII* construct along with a genomic DNA fragment of variable size (left border), and another fragment containing the 3′ region of the *AphVIII* construct with the corresponding genomic segment (right border). Following the digestion, the ligation of the DNA fragments was performed using T4 DNA ligase enzyme (Roche, Basilea, Switzerland) at 16 °C for 16 h. The ligated DNA served as a template for a PCR amplification using specific primers for the *AphVIII* cassette. To minimize non-specific amplifications, a nested PCR approach was used. The primers used for the first round were FL1 (CGCCGGGCTTGCTCGTC) and FL2 (CAGCGTGCTTGCAGATTTGACTTG) for the left border and FR1 (CCAGAGCTGCCACCTTGACA) and FR2 (AGCTGGCCCACGAGGAGGAC) for the right border. The primers used for the second round were SL1 (TCGGGGTCGCGGGCTTTTAT) and SL2 (CCCTCCCCGGTGCTGAAGAA) for the left border, and SR1 (CCACCACCCCGAAGCCGATAA) and SR2 (TACCGGCTGTTGGACGAGTTCTTCTG) for the right border. In the first round, 1 µL of the ligation mixture was used as the template DNA, and, in the second round, 1 µL of the product resulting from the first round was used as the template. In both rounds, a master mix reaction was prepared using a thermostable Taq polymerase (Biotools, Madrid, Spain), following the manufacturer’s protocol. The program used in each of the amplification rounds was as follows: for the first round, initial denaturation at 96 °C for 3 min; this was followed by 25 cycles of denaturation at 96 °C for 1 min, primer annealing at 60 °C for 1 min, and extension at 72 °C for 2 min; finally, there was an extension at 72 °C for 10 min. The conditions for the second round were the same as the first, except for increasing the number of PCR cycles to 40.

### 2.10. Bioinformatics Tools

The following databases and software tools were used in this study: Phytozome (https://phytozome-next.jgi.doe.gov/, accessed on 20 May 2025) for retrieving and analyzing genome sequences of *Chlamydomonas*, and NCBI BLAST (https://blast.ncbi.nlm.nih.gov/Blast.cgi, accessed on 20 May 2025) for nucleotide and protein sequence analysis. ClustalW (https://www.genome.jp/tools-bin/clustalw, accessed on 20 May 2025) and BioEdit software v7.2 were used for nucleotide and protein sequence alignments, respectively. The DNAStar (Lasergene, Madison, WI, USA) software suite (https://www.dnastar.com/, accessed on 20 May 2025) included SeqMan for managing and analyzing DNA sequencing data and PrimerSelect for primer design. MEME (Multiple Expectation Maximization for Motif Elicitation) (http://meme-suite.org/, accessed on 20 May 2025) was employed to identify conserved protein domains. Cello2go (http://cello.life.nctu.edu.tw/cello2go/, accessed on 20 May 2025) facilitated the prediction of protein subcellular localization, ProtParam (https://web.expasy.org/protparam/, accessed on 20 May 2025) was used to determine the physicochemical properties of proteins based on their sequences. BioRender (https://biorender.com/, accessed on 20 May 2025) was utilized for creating scientific illustrations.

### 2.11. Statistical Analysis

Data are presented as the mean ± standard deviation (SD) from at least three independent experiments. Error bars represent the SD. Statistical comparisons were performed using an unpaired, two-tailed Student’s *t*-test, with the parental strain 704 used as the reference group in all cases. Statistical significance was defined as follows: *p* < 0.05 (*), *p* < 0.01 (**), and *p* < 0.001 (***).

## 3. Results

### 3.1. Generation of an Insertional Mutant Collection

The *Chlamydomonas* parental strain selected was the strain 704. The key distinguishing feature of this strain, compared to wild-type *Chlamydomonas*, is that it has a mutation in the genes involved in cell wall formation. This mutation results in a thinner cell wall, enabling highly efficient transformation and the generation of a large number of transformants [30]. Aside from the cell wall mutation, strain 704 does not possess any additional mutations affecting Mo or nitrate homeostasis, making it an ideal choice for the objectives of this study. The DNA used for the transformation of strain 704 was the *AphVIII* gene, which confers resistance to the antibiotic paromomycin and serves as a selectable marker. The *AphVIII* gene was amplified by PCR from the plasmid pSI104 (Figure 1A). This marker has been successfully employed in other functional genomics strategies in *Chlamydomonas* [32]. This microalga lacks the ability for homologous recombination, so, during transformation, the genetic marker is inserted randomly into its genome [37]. This random insertion can have significant consequences. If the marker is inserted within a gene sequence, it can disrupt the gene’s structure and function, potentially causing a mutation.

A total of 150 transformations were performed using *AphVIII*, including several negative controls lacking DNA. One example of the transformation results is shown in Figure 2. On average, 20–50 transformants were obtained per selection plate, yielding approximately 5200 paromomycin-resistant transformants. The transformation efficiency achieved was approximately 3500 transformants per µg of DNA, which is highly comparable to the efficiencies reported in previous studies [38]. The transformants were sufficiently separated to allow for the individual isolation and long-term preservation of each one (Figure 2).

### 3.2. Phenotypic Analysis of Transformants in Relation to Mo and Nitrate Homeostasis

Each paromomycin-resistant colony was individually isolated and preserved; from this point onward, they are referred to as transformants. The transformants were named based on their position within the selection plate. To analyze potential phenotypes related to Mo homeostasis, the entire collection of 5200 transformants was spot-cultured on agar plates containing different Mo concentrations and N sources (Figure 1D). The conditions chosen for the phenotypic screening of the transformants were as follows: the *Chlamydomonas* culture medium with ammonium as N source and containing the standard concentration of 1 µM Mo (referred to as Ammonium); a medium identical to the previous one, supplemented with paromomycin to verify the presence of the *AphVIII* marker in the transformants (referred to as Ammonium-PAR). *Chlamydomonas* can grow in media using either ammonium or nitrate as a N source. However, Mo is essential only in nitrate-containing media, as it is required by the NR enzyme, which uses Mo as a cofactor. For this reason, nitrate was chosen as the N source in the culture media to analyze the effect of Mo on growth. Therefore, the transformants were screened in a medium where ammonium was replaced with nitrate, containing the same amount of Mo as the standard *Chlamydomonas* medium (referred to as Nitrate Normal-Mo). This condition also helps us to detect strains incapable of growing on nitrate. Additionally, two other media were used to further investigate Mo-related phenotypes: a medium with nitrate but prepared without adding Mo (referred to as Nitrate Low-Mo), used to study the efficiency of Mo assimilation under Low-Mo stress; and a medium with nitrate supplemented with 10 mM Mo (referred to as Nitrate High-Mo), used to examine Mo toxicity and tolerance. The parental strain 704, along with the 5200 transformants, was then cultivated on these plates for comparative analysis.

The growth of the parental strain 704 and all the transformants was observed and compared in these media. After analyzing the 5200 transformants, all except the parental strain 704 (Figure 3) were able to grow in Ammonium-PAR medium, confirming that all the transformants carry the *AphVIII* marker inserted into their genome. Four of these transformants exhibited phenotypic behaviors different from the parental strain 704. Figure 3 shows actual images of the spot test for the parental strain and these four transformants that display differential phenotypic behavior. As previously reported, the parental strain 704 was able to grow in nitrate medium with normal Mo concentrations [30]. The screening revealed that two transformants (81.90 and 8.2) were unable to grow in nitrate-containing media, regardless of the presence or absence of Mo. This suggests that the insertion of the *AphVIII* marker affected genes involved in the nitrate metabolism in these transformants. 

Initially, we thought that strain 704 would not be able to grow in nitrate media without Mo added (Nitrate Low-Mo), as this is required for NR. However, strain 704 exhibited reduced growth in Low-Mo media compared to its growth in Normal-Mo media, as indicated by the less intense green spot color. Nevertheless, it still managed to grow in the Low-Mo media (Figure 3). This suggests that, even in media prepared without intentionally added Mo, trace amounts of Mo must be present as an impurity in some of the reagents used to prepare the standard culture medium. That is why we have termed this medium without added Mo “Low-Mo”. Despite no intentional addition of Mo, it must contain trace amounts that allow for some growth. These trace amounts of Mo are sufficient for strain 704 to grow in nitrate-containing media; however, although the 704 strain can grow using trace amounts of Mo, its growth is significantly reduced compared to media containing the Normal-Mo concentration (1 µM) (Figure 3). This observation suggests that, while 704 can survive on minimal Mo levels, optimal growth requires Mo at higher concentrations. However, among the transformants that grew in Low-Mo nitrate media, none showed altered growth compared to the parental strain. Therefore, our initial idea of finding transformants with altered low Mo assimilation was not successful. In High-Mo medium (10 mM), the growth of the parental strain 704 was noticeably diminished, and two transformants (2.2 and 9.34) showed appreciable differences compared to the parental strain. Transformant 2.2 was more resistant to High-Mo concentrations, and transformant 9.34 was more sensitive to High-Mo than the parental 704 strain.

This initial screening on agar plates enabled us to perform a large-scale selection of transformants, identifying four strains as promising candidates. However, this visual assessment provides only qualitative results. Therefore, to confirm the phenotypes observed in the initial screening and to quantify growth more precisely, we cultivated the selected transformants in liquid medium, which allows for accurate chlorophyll quantification. The amount of chlorophyll is directly related to the growth of microalgae, allowing for the precise quantification of their growth. As shown in Figure 4, the growth of strain 704 and transformants 2.2 and 9.34 was approximately 40% lower in Low-Mo nitrate medium compared to Normal-Mo nitrate medium. Transformants 81.90 and 8.2 exhibited a complete inability to grow in nitrate-containing media, confirming the result obtained in the agar plate. Transformant 2.2 exhibited approximately 50% greater growth than the parental strain in Nitrate High-Mo medium, whereas mutant 9.34 showed notably lower growth under the same condition. Overall, the results from the liquid cultures confirmed the phenotypes observed in the initial screening.

### 3.3. Localization of the AphVIII Marker Insertion in the Transformants

After selecting the four transformants, the next step was to determine the genomic region disrupted by the insertion of the *AphVIII* marker. This is important because the gene in which the insertion occurred could be responsible for the observed phenotype of resistance or sensitivity to high Mo or inability to grow in nitrate. To localize the *AphVIII* marker insertion, a ligation-mediated PCR strategy was employed, based on the Inverse Polymerase Chain Reaction method [36]. This process involved the following steps (Figure 5): isolation of total genomic DNA from each mutant; digestion of the genomic DNA with the restriction endonuclease *AvaI*—this enzyme was chosen because it cuts only once within the *AphVIII* marker and frequently throughout the *Chlamydomonas* genome; ligation of the resulting DNA fragments to produce a large number of circular DNA molecules. Some of these circular DNA fragments should contain part of the *AphVIII* marker along with adjacent *Chlamydomonas* genomic DNA. The ligation products obtained were used as templates for a first round of PCR, followed by a second, nested PCR round using the product from the initial PCR as a template. The nested PCR approach enhances both the specificity and yield of the desired amplicons. For this purpose, specific *AphVIII* marker primers were employed to enable the amplification of both the right and left sides of the insertion site. Further details about the procedure are provided in Section 2. This strategy is particularly effective for amplifying unknown genomic sequences adjacent to the known *AphVIII* marker sequence.

The bands obtained after the PCR of the four transformants are shown in Figure 6. As can be seen, amplification failed for some primer pairs, while others were successful. However, in all transformants, the amplification of at least one border was achieved, allowing us to identify the genomic insertion site. All bands larger than 300 bp were excised from the agarose gel and sent for sequencing. By sequencing the PCR products, we were able to identify the genomic regions flanking the *AphVIII* insertion site, thus revealing the precise location of the marker in the *Chlamydomonas* genome and identifying potentially disrupted genes.

The sequencing results obtained were compared with the *Chlamydomonas* genome database (phytozome), considering as valid those sequences that presented a clear junction between the *AphVIII* and a known chromosomal region. Table 1 summarizes the results, indicating the chromosome, the gene locus, and the predicted function of the affected gene.

The table provides details on the transformant number, the chromosome where the insertion is located, the affected locus (gene ID), and the predicted function of the disrupted gene, including protein type.

Transformants 2.2 and 9.34 had the marker insertions in genes with homology to ABC-type transporters: *Cre05.g234400* in transformant 2.2 and *Cre03.g191350* in transformant 9.34. ABC-type transporters are a family of membrane proteins that have been implicated in the transport of various types of substrates across membranes. These substrates include a wide range of molecules. We have analyzed the genomic context surrounding these two genes in the *Chlamydomona*s genome. As shown in Figure 7, the genes flanking each type of ABC-type transporter do not, when their functions are known, appear to be involved in the transport of any solute. Therefore, in this case, chromosomal localization does not provide evidence to support their function. However, in bacteria, Mo is among the substrates known to be transported by certain ABC-type transporters [39].

In *Chlamydomonas*, 75 ABC-type transporters have recently been identified and classified into the following subfamilies based on phylogenetic analysis: 7 ABCA, 8 ABCB, 10 ABCC, 3 ABCD, 1 ABCE, 11 ABCF, 26 ABCG, and 9 ABC [40]. In this study, the protein encoded by the gene *Cre05.g234400* was classified as belonging to the ABCG subfamily, while the protein encoded by the gene *Cre03.g1913500* was assigned to the ABCA subfamily. Figure 8A shows the analysis of conserved domains in the protein encoded by the gene *Cre05.g234400*. Two characteristic domains of the ABC family are present. First, the ABC domain is located between residues 558 and 758, with an E-value of 1.3 × 10^−22^, indicating high statistical significance and suggesting strong conservation and functionality of this domain. Second, the ABC domain is located between residues 2203 and 2410, with an E-value of 3.9 × 10^−8^, also statistically significant, although less conserved than the first. The presence of two separate ABC domains suggests that the protein may have a typical structure of full ABC transporters, which usually contain two ATP-binding domains. This supports the functional annotation of the gene as an ABC-type transporter. Figure 8B illustrates the analysis of conserved domains in the protein encoded by the *Cre03.g191350* gene. Two well-defined ABC domains were identified. The first ABC domain was located between residues 984 and 1169, with an E-value of 6.1 × 10^−18^. This extremely low value indicates high statistical significance, suggesting a strong functional conservation of this domain. The second ABC domain was located between residues 3323 and 3467, with an E-value of 2.9 × 10^−14^, also highly significant and characteristic of ABC transporters. The considerable distance between both domains suggests the existence of intermediate regions, likely corresponding to transmembrane domains or spacer regions, which is common in this type of transporter. Their possible transmembrane topology and the subcellular localization of these possible transporters were studied using the TMHMM 2.0 and WoLF PSORT tools, respectively. The results indicated that both proteins have a high probability of being located in the plasma membrane. The protein encoded by the gene *Cre05.g234400* is predicted to have 4 possible transmembrane elements, while the protein encoded by the gene *Cre03.g191350* is predicted to have 14 transmembrane elements (Figure 8).

Regarding the transformants unable to grow on nitrate, in 81.90, the marker insertion occurred in the *Cre08.g382545* gene. Figure 8C shows the conserved domain analysis of the protein encoded by the *Cre08.g382545* gene. The most relevant finding is the identification of a ThiS domain located between residues 5 and 83 (E-value of 4.1 × 10^−23^), indicating very high statistical significance and a strong evolutionary conservation of this domain. The ThiS domain is characteristic of proteins involved in Moco biosynthesis. The domain spans nearly the entire protein (residues 5 to 83 out of a total of 84), suggesting that the main function of the protein is directly associated with this domain. The protein encoded by this gene shares 40.1% homology with CNX7 from *Arabidopsis thaliana*, suggesting a similar function. CNX7 is crucial for the biosynthesis of Moco, and this mutation accounts for the mutant’s inability to grow on nitrate as the sole N source. Without a functional *CNX7* gene, the strain cannot produce the Moco necessary for activating NR, the key enzyme in nitrate assimilation. As a result, mutant 81.90 should lack active NR, making it incapable of utilizing nitrate for growth.

In transformant 8.2, the marker was inserted in the gene *Cre07.g348040*. Notably, the protein encoded by this gene shares 61% homology with the amino acid permease BAT1 found in Arabidopsis [41], suggesting a potentially similar function. The possible transmembrane topology and subcellular localization of BAT1 were analyzed using the TMHMM 2.0 and WoLF PSORT tools, respectively. The results predict that BAT1 contains 11 transmembrane domains and has a high probability of being localized to the plasma membrane. Figure 8D presents the conserved domain analysis of the protein encoded by *Cre07.g348040*, annotated as a BAT1-type amino acid permease. The most relevant finding is the identification of an AA permease 2 domain, located between residues 11 and 480, with an E-value of 1.2 × 10^−29^, indicating very high statistical significance and a strong evolutionary conservation of this domain. The AA permease 2 domain is characteristic of proteins that transport amino acids across the plasma membrane. The presence of this domain in the BAT1 protein suggests that it plays an essential role in amino acid uptake and exchange. Transformant 8.2 presents an interesting case in our study. Its inability to grow on nitrate is intriguing, given that, as far as is known, BAT1 is not directly involved in nitrate assimilation, unlike CNX7. In the Section 4, we propose possible explanations for this unexpected phenomenon.

## 4. Discussion

Mo homeostasis includes aspects that remain unknown, such as Mo efflux transport (critical for long-distance transport), mechanisms of Mo storage (necessary to provide a rapid Mo source under conditions of high demand), and cellular strategies for Mo sensing and regulation (essential for coordinating Mo availability with cellular demand) [42]. Identifying the molecular players involved in these processes is crucial for improving N assimilation, as NR, a key enzyme in N metabolism, depends on Mo [17]. Therefore, a comprehensive understanding of Mo homeostasis is critical not only to meet the increased metal demand associated with N assimilation but also to prevent potential Mo deficiencies that could impair metabolic function.

Our working hypothesis is that *Chlamydomonas* has homeostatic mechanisms to regulate intracellular Mo concentration, and that these mechanisms are shared by higher plants. Supporting this hypothesis is the fact that much of the current knowledge on plant Mo homeostasis has been derived, directly or indirectly, from studies on *Chlamydomonas* [43]. The success of this model has been previously demonstrated, as the two known families of Mo-specific transporters in eukaryotes (MOT1 and MOT2) were first identified in *Chlamydomonas* [13,14]. Both MOT1 and MOT2 belong to the Major Facilitator Superfamily (MFS), a large group of membrane transport proteins that facilitate the movement of small solutes across cell membranes. MOT1 is present not only in algae but also in higher plants and fungi, where it mediates high-affinity Mo uptake [13]. In *Arabidopsis thaliana*, MOT1 transporters have been reported to play a role in efficient Mo uptake from the soil [44], with their absence leading to impaired plant growth [45]. Members of the MOT1 family have also been identified as specific Mo transporters in other eukaryotic organisms, such as *Medicago truncatula* and rice [46,47,48]. MOT2 was also first identified in *Chlamydomonas*, and subsequent research using heterologous expression has suggested its involvement in Mo transport in humans [14].

We conducted a screening in nitrate-containing medium under three different Mo concentrations: normal concentration (1 μM), no added Mo (Low-Mo), and High-Mo (10 mM). Initially, we hypothesized that *Chlamydomonas* would not grow in the absence of added Mo. However, it did show growth, albeit reduced (Figure 3 and Figure 4). This observation would imply that, even when Mo is not intentionally added to the culture medium, its existing concentration is sufficient to sustain *Chlamydomonas* growth. This suggests that trace amounts of Mo may originate from impurities in other medium components, such as sulfate or phosphate salts, which are structural analogs of molybdate [49,50]. One of the aims of this screening was to identify genes involved in Mo homeostasis under limiting conditions. From this perspective, the screening with Low-Mo was not successful.

In contrast, the High-Mo conditions yielded more promising results. We used 10 mM Mo, a concentration previously known to be toxic in *Chlamydomonas* [51], with the aim of identifying transformants that were either more resistant or more sensitive than the parental strain 704 to these high concentrations. The objective was to identify genes responsible for regulating and detoxifying excess Mo. There are several potential reasons for the toxicity of high Mo concentrations. In general, elevated metal concentrations can lead to the generation of reactive oxygen species through Fenton-type reactions, resulting in damage to cellular components [52]. Moreover, Mo acts as a copper antagonist [53], and excessive Mo levels can disrupt copper homeostasis. Such disruption has been linked to disorders like Menkes and Wilson diseases, characterized by copper deficiency and overload, respectively [54]. In plants, since copper is essential for photosynthesis [55], imbalances caused by high Mo levels could impair this process. The antagonistic relationship between these two metals has been proposed as a key factor in their metabolism. It has been suggested that copper interferes with Moco biosynthesis by inhibiting the magnesium-dependent Mo insertion reaction catalyzed by CNX1G [56]. Another possible explanation for the toxic effect of high Mo concentrations is its chemical similarity to other oxyanions, particularly sulfate and phosphate. Molybdate can be non-specifically taken up via sulfate transport systems, as reported for SHST1, a high-affinity sulfate transporter from *Stylosanthes hamata* which facilitates Mo uptake [57], illustrating the overlap in transport mechanisms between these two ions. Another possible explanation for the toxic effect of high concentrations of Mo could be related to phosphate. The phosphate transport system may also play a role in the non-specific absorption of molybdate. Molybdate and phosphate are highly similar in size, charge, and tetrahedral structure, leading to potential overlap in their transport mechanisms [58]. This is evidenced by observations in tomato plants, where a deficiency in phosphate leads to an increased uptake of molybdate, suggesting cross-talk between these transporter systems [59].

Apart from MOT1 and MOT2, no other Mo transport systems have been identified in eukaryotic organisms. However, more transport systems could exist, particularly those involved in exporting Mo out of the cell to prevent toxic accumulation. Two of the genes identified in this study, *Cre05.g234400* and *Cre03.g191350*, encode ABC-type transport proteins, which may represent novel mechanisms for regulating Mo transport. ABC transporters are essential components of cellular homeostasis and survival across all life forms due to their ability to handle a wide range of substrates with high specificity and efficiency [60]. These substrates include nutrients, metabolites, peptides, metals, ions, and amino acids, which are moved across cellular membranes into or out of the cytoplasm. Their main characteristic is a common architecture consisting of transmembrane domains (TMDs) and nucleotide-binding domains (NBDs). TMDs are embedded in the membrane and are responsible for substrate recognition and translocation. ABC transporters operate via an active transport, requiring ATP hydrolysis to move substrates against their concentration gradient [61]. Interestingly, two families of ABC-type transporters have been identified in bacteria involved in Mo transport: the MolBC-A and the ModABC system. The MolBC-A functions as a low-affinity system and is active during periods of high extracellular Mo concentration [62]. It works by transferring Mo from the periplasmic binding protein MolA to the transmembrane domain MolB. In contrast, the ModABC system acts as a high-affinity Mo transporter and includes ModA, a periplasmic binding protein; ModB, the membrane-spanning domain; and ModC, the ATPase that powers the transport process [39].

The ABC-type transporters identified could be involved in either importing Mo into the cell or exporting it to the extracellular environment. Among the transformants analyzed, mutant 2.2, which carries a mutation in the gene *cre05.g234400*, exhibited higher resistance to elevated Mo concentrations compared to the parental strain. This phenotype suggests that the ABC transporter encoded by *cre05.g234400* (Cre05) may be involved in Mo uptake or storage (Figure 9). Its disruption could reduce Mo entry into the cell or promote the enhanced sequestration of excess Mo, thereby conferring increased tolerance to high Mo concentrations. In contrast, the transformant 9.34, which carries a mutation in the gene *Cre03.g191350*, displayed lower tolerance to elevated Mo concentrations compared to the parental strain. This observation suggests a potential defect in genes involved in Mo efflux mechanisms. If the ABC transporter encoded by *Cre03.g191350* (Cre03) is involved in Mo export, its mutation could impair the cell’s ability to eliminate excess Mo, leading to toxic accumulation and increased sensitivity. Consequently, we hypothesize that *Cre03.g191350* encodes a transporter potentially involved in Mo efflux (Figure 9).

Once Mo enters the cell, it can be either immediately utilized for Moco synthesis or stored to ensure a reserve for future metabolic needs. In bacteria, two distinct Mo storage systems have been identified: one involving molybdate binding to Molbindin-type proteins, and another utilizing MoSto proteins [63,64]. However, no homologs of these bacterial storage proteins have been found in eukaryotes. Our study did not identify any proteins with homology to known Mo storage components, suggesting that eukaryotes may rely on alternative strategies for Mo storage or may have a reduced requirement for long-term Mo reserves. Before being inserted into molybdoenzymes, Mo must be coordinated with MPT to become biologically active in eukaryotes and to form Moco [22]. Moco is synthesized through a highly conserved pathway, from bacteria to mammals [65] (Figure 9). The first step of Moco biosynthesis involves the conversion of Guanosine Triphosphate (GTP) into cyclic pyranopterin monophosphate (cPMP), a reaction catalyzed by the proteins CNX2 and CNX3 [66]. The second step involves the transfer of two sulfur atoms to cPMP, forming MPT through catalysis by the MPT synthase complex, which consists of two small CNX7 subunits and two large CNX6 subunits [11]. Subsequently, the MPT synthase must be re-sulfurated by the MPT synthase sulfurase CNX5 [67]. In the third step, MPT is activated by adenylation catalyzed by CNX1G [68,69]. In the final step, MPT is deadenylated by CNX1E, a process that enables the insertion of Mo into MPT, resulting in the formation of active Moco [70]. After its synthesis, Moco is transferred to the different molybdoenzymes, such as NR [71,72].

Although the main goal of this work was to identify genes involved in Mo homeostasis, the screening strategy also led to additional findings. Transformants resistant to paromomycin were initially selected in ammonium-containing medium, and their Mo homeostasis-related phenotypes were later evaluated in nitrate-containing medium. This two-step process enabled the identification of mutants with defects not only in Mo homeostasis but also in nitrate utilization. Specifically, we identified two mutants incapable of growing on nitrate regardless of Mo availability. One of them harbored a mutation in the gene *Cre08.g382545*, which encodes CNX7 protein, the small subunit of the MPT synthase. This phenotype is consistent, as, without CNX7, the synthesis of an active Moco cannot occur, NR cannot function properly, and growth on nitrate is not possible. We are uncertain why, out of the 5200 mutants analyzed, only mutant 81.90 harbors a mutation in a *CNX* gene—specifically *CNX7*—and why no transformants were identified in other *CNX* genes. This result could be due to random chance, or it may reflect the essential roles that Moco might play in pathways beyond nitrate metabolism. While Moco is known to be critical for nitrate assimilation via enzymes such as NR, it is also involved in ammonium metabolism, where, as mentioned, other molybdoenzymes such as sulfite oxidase, aldehyde oxidase, and xanthine dehydrogenase play key roles [15]. The broader metabolic relevance of Moco, including in ammonium metabolism, could cause Moco-deficient mutants to exhibit slower growth rates and delayed colony formation. Consequently, such slow-growing mutants may have been overlooked during our initial screening, which was performed in ammonium-containing media supplemented with paromomycin (Figure 2). This remarkably low recovery of mutants with alterations in *CNX* genes suggests that disrupting these genes is particularly challenging. In this sense, a previous mutant library screening aimed at identifying genes involved in nitrate assimilation recovered only a single mutant in the *CNX2* gene among 22,000 transformants isolated [73].

Mutant 8.2 presents an intriguing case in our study, as the marker is inserted in the gene *Cre07.g348040*, which encodes a protein that shares similarities with the bidirectional amino acid permease BAT1. This mutant exhibits an inability to grow on nitrate, which is particularly interesting since the affected gene is not directly part of the nitrate assimilation pathway. In *A. thaliana*, BAT1 is notable for its dual transport activity—both the export and import of amino acids—hence its name: Bidirectional Amino Acid Transporter 1 [41]. BAT1 belongs to a family of integral membrane proteins that transport amino acids across cell membranes and contains conserved domains common to amino acid transporters found in both prokaryotes and eukaryotes, suggesting that it is part of an evolutionary ancient family of transporters preceding the divergence of these two domains of life [74]. BAT1 is essential for the uptake of amino acids, which serve as organic N sources. However, its connection to the nitrate-dependent phenotype of mutant 8.2 is not immediately clear, given that nitrate is an inorganic N source. This unexpected phenotype suggests a possible link between amino acid transport and nitrate metabolism, or, alternatively, an unknown function of BAT that is essential for nitrate utilization (Figure 9). In plants, nitrate is absorbed from the soil and reduced to nitrite and subsequently to ammonium, which is then incorporated into amino acids, primarily glutamate [6]. Glutamate is not only the most abundant amino acid in many plant species, but also the first organic N compound formed in this pathway. It acts both as a crucial metabolic intermediate and as a signaling molecule involved in the regulation of N metabolism [75]. It has been suggested that the internal amino acid pools, particularly glutamate, can act as a signal of N status, modulating nitrate uptake and transporter expression. In support of this idea, exogenously applied amino acids, particularly glutamate, have been shown to decrease nitrate influx and downregulate transporter transcription in root tissue [76]. This regulatory mechanism may occur both at the transcriptional level and through post-translational modification. BAT1 has been reported to transport several amino acids, including alanine, arginine, glutamate, and lysine, with a particularly high affinity for glutamine, and its expression is strongly induced in the presence of glutamine [77]. Moreover, recent studies indicate that glutamine availability directly influences the expression of key nitrate transporters, such as NRT2.4, which are vital for nitrate-dependent growth [78]. Specifically, glutamine depletion has been observed to inhibit the expression of these transporters, suggesting that glutamine acts as an internal signal that helps plants to fine-tune their N uptake mechanisms based on their current N status. Based on this evidence, we hypothesize that mutant 8.2 may have an altered transport mechanism for an essential amino acid, most likely glutamine or glutamate, which could impair its ability to properly regulate nitrate metabolism. This would explain its inability to grow efficiently on nitrate as the sole N source. Further experimental validation is required to verify this hypothesis and to determine the precise role of BAT1 in nitrate-related pathways. Regarding our future research on Mo homeostasis, we will focus on characterizing Mo transport activity in these mutants compared to the wild type. This approach will help to elucidate the specific roles of these ABC transporters in Mo metabolism. Additionally, another strategy we plan to implement to identify new agents involved in Mo homeostasis is to perform transcriptome analyses under different experimental conditions, using varying concentrations of Mo.

## 5. Conclusions

Through an extensive insertional mutagenesis screening, this study has identified novel candidate genes involved in Mo and nitrate homeostasis in *Chlamydomonas*. Among the most relevant findings, we uncovered two ABC transporter family genes potentially involved in Mo transport, one gene essential for the Moco biosynthesis, and the *BAT1* gene, a putative bidirectional amino acid transporter with a possible regulatory role in nitrate metabolism. These findings expand our understanding of the genetic network involved in Mo uptake, cofactor biosynthesis, and N assimilation. The identification of these genes provides a solid foundation for future investigations aimed at elucidating their specific biological functions and interactions in Mo and nitrate metabolism. Moreover, our findings contribute to a better understanding of nutrient uptake and utilization, the regulation of Mo-dependent processes, and the connection between amino acid transport and nitrate assimilation. The insights gained have potential implications for both agricultural and environmental applications, advancing our knowledge in the broader field of nutrient metabolism.

## Figures and Tables

**Figure 1 cimb-47-00396-f001:**
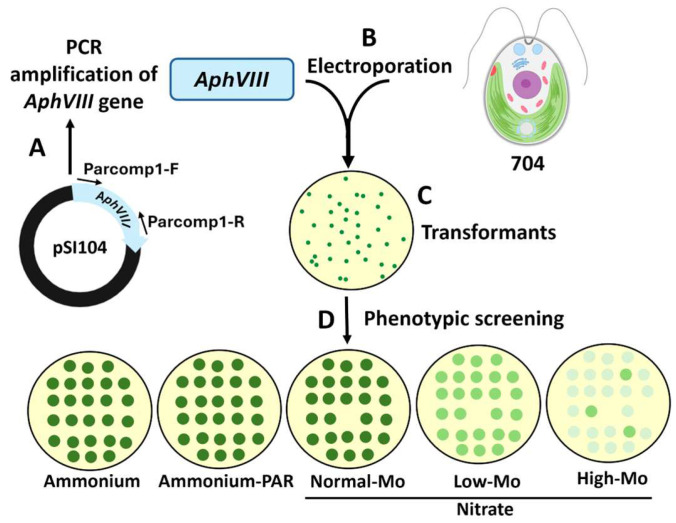
Schematic representation of the transformation process via insertional mutagenesis and selection of the transformants. (**A**) The *AphVIII* marker gene was amplified by PCR from the vector pSI104. Parcomp1-F and Parcomp1-R are the primers used for the PCR amplification of *AphVIII*. (**B**) The *Chlamydomonas* strain 704 was transformed with *AphVIII* by electroporation. (**C**) Illustration of a culture plate with paromomycin 25 µg/mL (PAR) in which transformants resistant to paromomycin can be observed. (**D**) Each of the transformants was grown individually, afterward the growth was analyzed using a drop test on culture plates with the indicated concentrations of ammonium or nitrate and Mo. Ammonium (8 mM ammonium and 1 µM Mo), Ammonium-PAR (8 mM ammonium, 1 µM Mo, and 25 µg/mL paromomycin), Normal-Mo (4 mM nitrate and 1 µM Mo), Low-Mo (4 mM nitrate and no added Mo), High-Mo (4 mM nitrate and 10 mM Mo).

**Figure 2 cimb-47-00396-f002:**
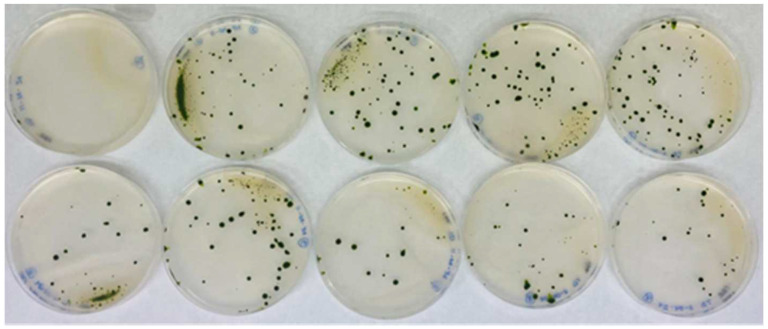
Representative example of transformation plates obtained from the same batch. The figure shows the results of nine transformations plus a control from the 150 transformations that were performed. The top-left plate shows the control plate, in which the transformation was carried out without the *AphVIII* marker DNA. The other plates included DNA from the *AphVIII* marker gene in the transformation. The culture medium of the plates was Ammonium-PAR (8 mM ammonium, 1 µM Mo, and 25 µg/mL paromomycin). Each of these plates is an actual image corresponding to what is schematically illustrated in Figure 1C.

**Figure 3 cimb-47-00396-f003:**
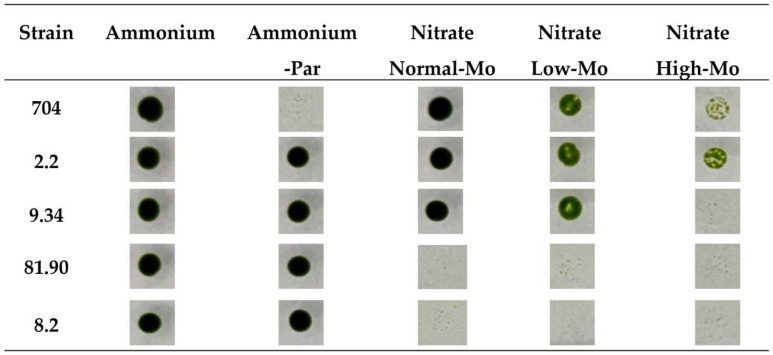
Comparison of the growth of the parental *Chlamydomonas* strain 704 and four selected transformants on agar plates. Approximately 500 cells from each of the indicated strains were spotted on the media specified at the top of the figure. The media compositions were as follows: Ammonium (8 mM ammonium and 1 µM Mo), Ammonium-PAR (8 mM ammonium, 1 µM Mo, and 25 µg/mL paromomycin), Normal-Mo (4 mM nitrate and 1 µM Mo), Low-Mo (4 mM nitrate and no added Mo), High-Mo (4 mM nitrate and 10 mM Mo). All culture media contained 1.6% agar. The plates were incubated in the growth chamber for 7 days. Afterward, the plates were photographed, and the results are shown. The data presented are representative of three biologically independent replicates.

**Figure 4 cimb-47-00396-f004:**
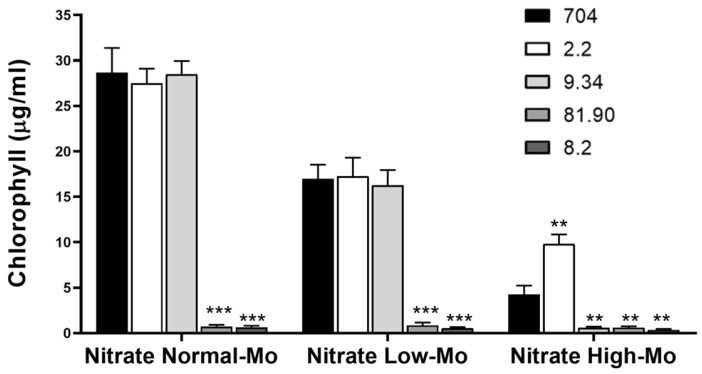
Comparison of the growth of the parental *Chlamydomonas* strain 704 and four selected transformants in liquid media. The indicated *Chlamydomonas* strains were inoculated in the specified liquid media at an initial concentration of 50,000 cells/mL. The media compositions were as follows: Nitrate Normal-Mo (4 mM nitrate and 1 µM Mo), Nitrate Low-Mo (4 mM nitrate and no added Mo), Nitrate High-Mo (4 mM nitrate and 10 mM Mo). Chlorophyll content was quantified after 7 days of growth as a measure of biomass accumulation. Error bars represent the standard deviation (SD) of three biological replicates. Statistical comparisons were performed against the corresponding parental strain 704. Statistical significance is indicated as follows: *p* < 0.01 (**), *p* < 0.001 (***).

**Figure 5 cimb-47-00396-f005:**
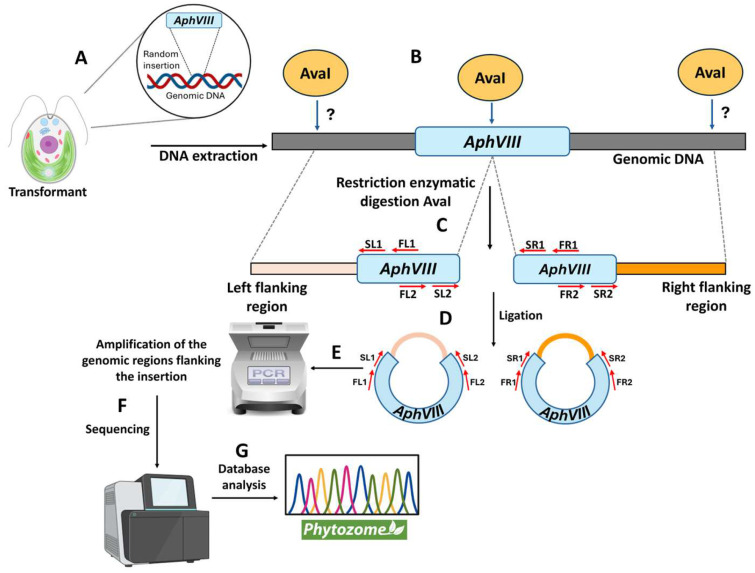
Schematic representation of the procedure to identify the genomic region flanking the *AphVIII* marker insertion. This figure illustrates the different stages of the process for identifying the genomic insertion site of the *AphVIII* marker. (**A**) *Chlamydomonas* was transformed with the *AphVIII* marker. This DNA fragment randomly inserts into an unknown location in the *Chlamydomonas* genome. Since the insertion site varies, the position of the genomic *AvaI* sites is unpredictable (indicated by a question mark). (**B**) Genomic DNA was isolated and digested with the restriction enzyme *AvaI*. (**C**) The left side of the insertion was considered the genomic region adjacent to the 5′ end of *AphVIII*, while the right side corresponded to the genomic region adjacent to the 3′ end of *AphVIII*. (**D**) The fragments resulting from the digestion were ligated. (**E**) The ligation DNA was used as a template for PCR amplification using the indicated primer sets (FL1, FL2, FR1, FR2, SL1, SL2, SR1, and SR2). (**F**) After the purification from agarose gel, the PCR products were sequenced. (**G**) The sequences obtained were analyzed using the *Chlamydomonas* Phytozome. For more specific details, see Section 2.

**Figure 6 cimb-47-00396-f006:**
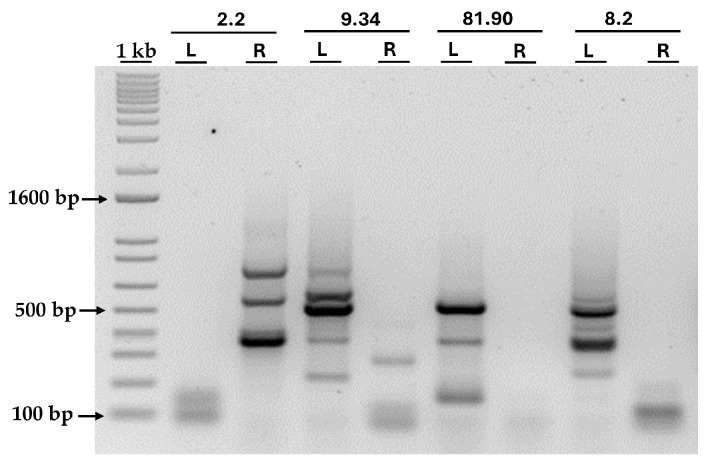
PCR amplification of the genomic DNA adjacent to the *AphVIII* insertion. The figure shows the agarose gel after loading the products from the second round of PCR amplification of the genomic DNA adjacent to the *AphVIII* insertion in the four selected transformants. Above the lanes, the names of the transformants and the primer pairs used are indicated: L (primers pair for the left border) and R (primers pair for the right border). More details about the procedure can be found in the Section 2. “1 kb” represents the molecular weight marker, and, next to it, the size in base pairs (bp) of three representative bands is indicated.

**Figure 7 cimb-47-00396-f007:**
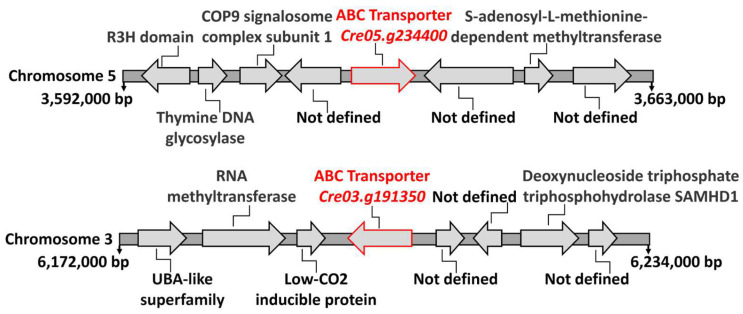
The genomic region surrounding the ABC-type transporters. This figure schematically shows the positions and annotations of several gene loci on the *Chlamydomonas* chromosomes where the identified ABC-type transporters are located. The information was retrieved from the *Chlamydomonas* Phytozome database (https://phytozome-next.jgi.doe.gov/info/Creinhardtii_v5_6, accessed on 20 May 2020). The positions at the beginning and end of each chromosome segment are indicated in base pairs (bp). The putative functions of each gene are annotated where available; “Not defined” indicates genes with unknown or uncharacterized functions. Arrows indicate the orientation of each gene.

**Figure 8 cimb-47-00396-f008:**
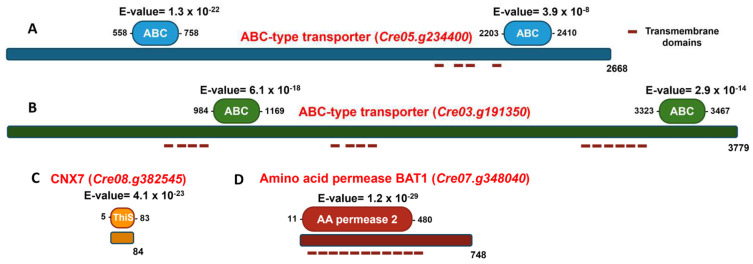
Schematic representation of the domain organization of ABC-type transporters, CNX7 and BAT1. The proteins are depicted as horizontal bars, with their lengths corresponding to the full amino acid sequences, indicated at the ends of the bars. Domains are shown as colored boxes, with their respective E-values displayed above each domain and the amino acid positions marking the start and end of each domain indicated accordingly. Red dashed lines represent putative transmembrane domains. (**A**) The ABC-type transporter encoded by *Cre05.g234400*, with its two conserved ABC domains highlighted in blue. (**B**) The ABC-type transporter encoded by *Cre03.g191350*, with its two conserved ABC domains highlighted in green. (**C**) The CNX7 protein encoded by *Cre08.g382545*, with its conserved ThiS domain highlighted in orange. (**D**) The amino acid permease BAT1 encoded by *Cre07.g348040*, with its conserved AA permease 2 domain highlighted in red.

**Figure 9 cimb-47-00396-f009:**
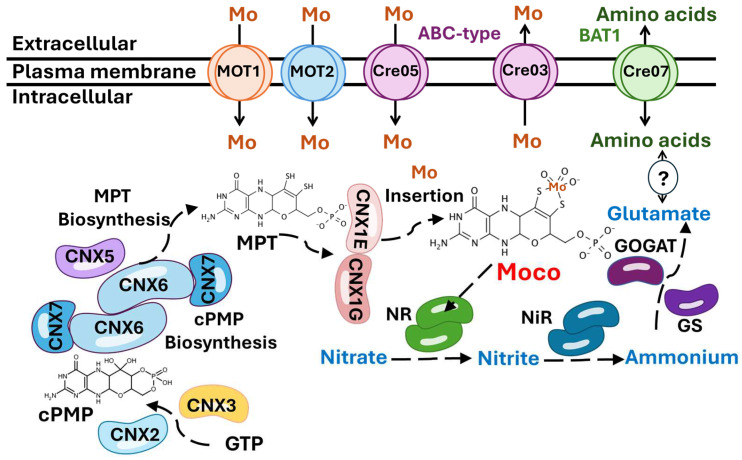
Schematic model of Mo and nitrate metabolism in *Chlamydomonas.* Proteins catalyzing individual steps are shown in different colors. All known intermediates of the pathway are presented sequentially in the three steps in which Moco is synthesized. CNX2 and CNX3 catalyze the conversion of GTP to cPMP. CNX5, CNX6, and CNX7 convert cPMP to MPT. MOT1 and MOT2 transport Mo to the cytosol. CNX1G and CNX1E incorporate Mo into MPT to produce Moco. Moco binds to nitrate reductase (NR), which catalyzes the reduction of nitrate to nitrite. Then, nitrite reductase (NiR) reduces nitrite to ammonium, and the Glutamine Synthetase/Glutamate Synthase (GS/GOGAT) enzymes convert ammonium into glutamate. The results obtained allow us to propose the hypothesis that the ABC transporter encoded by *Cre05.g234400* (Cre05) may be involved in the uptake or storage of Mo. In contrast, the ABC transporter encoded by *Cre03.g191350* (Cre03) may play a role in the export of Mo. Additionally, we hypothesize that there may be a connection between *Cre07.g348040* (Cre07), which encodes a protein homologous to the amino acid permease BAT1, and nitrate assimilation through the modulation of amino acid pools, as indicated by the question mark (?). For a more detailed explanation, refer to the text.

**Table 1 cimb-47-00396-t001:** Characterization of *Chlamydomonas* transformants identified in this study.

Transformants	Chromosome	Locus Affected	Predicted Function (Protein Type)
2.2	5	cre05.g234400	Solute transport (ABC transport)
9.34	3	cre03.g191350	Solute transport (ABC transport)
81.90	8	cre08.g382545	Moco synthesis (CNX7)
8.2	7	cre07.g348040	Amino acid transporter (BAT1)

## Data Availability

All data required to evaluate the conclusions of this paper are included in the main text.

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
