# Peer review of "Insertional Mutagenesis as a Strategy to Open New Paths in Microalgal Molybdenum and Nitrate Homeostasis"

_cimb, 2025, doi:10.3390/cimb47060396_

Round 1
Reviewer 1 Report
Comments and Suggestions for Authors
The study is comprehensive. A revision is suggested for this article. Several suggestions are attached below:
- As there are almost 50 ABC transporter members of this family, what specific ABC transporters are involved in the present investigation topic?
- Several errors were found, such as line 146, “Cells concentration was determined”; line 300, a space typing error; line 426, a space typing error. Please check the whole article again to correct all typing and grammar errors.
- Regarding Figure 4, please do the statistical calculations and mark the significance.
- The arrangement of figures is quite confusing; please move Figure 1 to the method part.
- The introduction is too lengthy. If possible, please try to streamline the contents.
Author Response
The authors would like to express our sincere gratitude to the reviewer for the time and effort devoted to evaluating our article. We believe that all the suggestions provided have significantly contributed to enhancing the clarity and quality of the manuscript. We have made every effort to address each of your comments thoroughly and accurately. We hope that our responses have satisfactorily resolved your concerns.
The study is comprehensive. A revision is suggested for this article. Several suggestions are attached below:
- As there are almost 50 ABC transporter members of this family, what specific ABC transporters are involved in the present investigation topic?
Yes, we agree. The phylogenetic analysis of all the Chlamydomonas ABC transporter family members was conducted two years ago and published in Marine Drugs. According to the authors of that study, the two transporters identified in our work belong to the ABCA and ABCG subfamilies. To clarify this point, we have added the new figure 8 and the following text to the manuscript.
L477: In Chlamydomonas, 75 ABC-type transporters have recently been identified and classified into the following subfamilies based on phylogenetic analysis: 7 ABCA, 8 ABCB, 10 ABCC, 3 ABCD, 1 ABCE, 11 ABCF, 26 ABCG, and 9 ABC [40]. In this study, the protein encoded by the gene Cre05.g234400 was classified as belonging to the ABCG subfamily, while the protein encoded by the gene Cre03.g1913500 was assigned to the ABCA subfamily. Figure 8A shows the analysis of conserved domains in the protein encoded by the gene Cre05.g234400. Two characteristic domains of the ABC family are present. First ABC domain: located between residues 558 and 758, with an E-value of 1.3e-22, indicating high statistical significance and suggesting strong conservation and functionality of this domain. Second ABC domain: located between residues 2203 and 2410, with an E-value of 3.9e-8, also statistically significant, although less conserved than the first. The presence of two separate ABC domains suggests that the protein may have a typical structure of full ABC transporters, which usually contain two ATP-binding domains. This supports the functional annotation of the gene as an ABC-type transporter. Figure 8B illustrates the analysis of conserved domains in the protein encoded by the Cre03.g191350 gene. Two well-defined ABC domains were identified: First ABC domain: located between residues 984 and 1169, with an E-value of 6.1e-18. This extremely low value indicates high statistical significance, suggesting strong functional conservation of this domain. Second ABC domain: located between residues 3323 and 3467, with an E-value of 2.9e-14, also highly significant and characteristic of ABC transporters. The considerable distance between both domains suggests the existence of intermediate regions, likely corresponding to transmembrane domains or spacer regions, which is common in this type of transporter.
- Li, X.; Li, X.; Yang, X.; Lan, C.; Huang, Y.; Jia, B. Identification and Characterization of ATP-Binding Cassette Transporters in Chlamydomonas Reinhardtii. Mar. Drugs 2022, 20, 63, doi:10.3390/md20100603.
- Several errors were found, such as:
-line 146, “Cells concentration was determined”; Fixed: Cell concentration was determined
-line 300, a space typing error; Fixed
-line 426, a space typing error. Fixed
-Please check the whole article again to correct all typing and grammar errors.
Fixed
- Regarding Figure 4, please do the statistical calculations and mark the significance.
Thank you for your suggestion. We have conducted the requested statistical analysis, and the results can be seen in Figure 4 and in the Materials and Methods section.
Please see Fig. 4 and_
2.11. Statistical analysis
Data are presented as the mean ± standard deviation (SD) from at least three independent experiments. Error bars represent the SD. Statistical comparisons were performed using an un-paired, two-tailed Student’s t-test, with the parental strain 704 used as the reference group in all cases. Statistical significance was defined as follows: p < 0.05 (*), p < 0.01 (**), and p < 0.001 (***).
- The arrangement of figures is quite confusing; please move Figure 1 to the method part.
We agree, done.
- The introduction is too lengthy. If possible, please try to streamline the contents.
We agree. We have tried to further condense the introduction, and these are the parts of the introduction that have been removed:
This organism can undergo both asexual and sexual reproduction, facilitating genetic studies (9).
Additionally, all these processes must be regulated and coordinated to match the biological demand for the nutrient and its availability
They observed that plants grown in purified nutrient solutions—designed to remove metal contaminants—developed deficiency symptoms leading to impaired growth. However, when these plants were treated with a diluted solution of molybdic acid, the deficiency symptoms disappeared, allowing for recovery and the resumption of normal growth (12).
In prokaryotes, Mo in the form of iron-molybdenum cofactor (FeMo-co) is essential for nitrogenase, the key enzyme involved in biological nitrogen fixation [16].
Accompanied by the transfer of an oxygen atom, either derived from or incorporated into water, this process changes the oxidation state of Mo from IV to VI (18).
This enzyme functions as a dimer, each monomer containing three domains: an N-terminal Fe₂-S₂ cluster, and FAD domain, and a C-terminal Moco domain [20].
In plants, the proteins involved in Moco synthesis are designated as CNX.
To carry out its catalytic activity, the mARC protein requires other associated proteins as electron donors [24].
Its structure varies across organisms: animal SO, found in mitochondria, has both heme and Moco domains; plant SO, located in peroxisomes, only has the Moco domain; Chlamydomonas possesses a SO with both domains (25)
With both showing some sequence similarity and prosthetic groups (28). However, AO derives from a duplication of the XDH gene [29].

Reviewer 2 Report
Comments and Suggestions for Authors
Reviewer’s comments
Title: Insertional Mutagenesis as a Strategy to Open New Paths in Microalgal Molybdenum and Nitrate Homeostasis
Manuscript Number: cimb-3634469
Journal: Current Issues in Molecular Biology
The research work entitled “Insertional Mutagenesis as a Strategy to Open New Paths in Microalgal Molybdenum and Nitrate Homeostasis” presents research on exploration of molybdenum uptake and nitrogen homeostasis pathways in green algae via insertional mutagenesis. Using this strategy, they identified 4 candidate genes from 5,200 mutants that may be involved in molybdenum transport, cofactor biosynthesis, and amino acid uptake. However, there are several concepts and diagrams proposed by the author in the article that needs to be properly modified, especially the graph in Fig.3 has repeated arrangement. Specific comments and general comments are given below:
Specific comments
- The logical flow of the Introduction appears somewhat disorganized. It is recommended to substantially revise this section by following a clearer structure: starting with the industrial significance of Chlamydomonas, followed by the importance of nitrogen uptake pathways in photosynthetic organisms, the role of molybdenum in these pathways, the current challenges in understanding molybdenum and nitrate homeostasis, and finally, the advantages of using the insertional mutagenesis strategy to address these gaps.
- Line 133-134, It is recommended to indicate the source of algae strains.
- In Figure 3, the image in the second column for strain 704 (Ammonium-Par) appears to be identical to the image in the third column for strain 81.90 (normal-Mo). This is likely an unintentional oversight during figure preparation. The authors are requested to review the original data and replace the duplicated image with the correct one.
- Line 528-534, Since the composition of Tris-Acetate-Phosphate (TAP) medium is publicly available, it is recommended that the authors clearly state that even 100 nM of molybdenum is sufficient to support green algal growth. It would also be advisable to avoid describing this trace metal uptake characteristic using the term "contaminants," as it may lead to misunderstanding. A careful revision of this paragraph is suggested for improved clarity and accuracy.
- Line 561-581, The authors propose that Cre05.g234400 and Cre03.g191350 may function in molybdenum ion transport, citing the presence of specific characteristic domains. To further support this claim, it is suggested that the authors include sequence alignment results—perhaps as Figure 7—for visual reference. A similar recommendation applies to Cre08.g382545 and Cre08.g348040, as such data would strengthen the overall argument.
- Since the molybdenum and nitrate homeostasis pathway discussed in this article is a complex pathway, it is recommended that the author provide a schematic diagram to improve readability.
General comments
- In the Materials and Methods section, the references to manufacturers are not formatted correctly. The proper citation format should be: (instrument model, manufacturer name, city, (state), country). For example, Line 160, (Microcellcounter F-500, Sysmex) à (Microcellcounter F-500, Sysmex, city, country). Other corrections required include line 180 (DNA Polymerase), line 196 (Gene Pulser), line 224 (NanoDrop), line 237 (restriction enzyme)
- Line 186 and 339, I would suggest revising the sentence to avoid using the future tense.
- Line 207, 50 mM Tris-HCl, pH 8.0; 5 mM EDTA; 300 mM NaCl à 50 mM Tris-HCl, pH 8.0, 5 mM EDTA, 300 mM NaCl (semicolon à comma)
- Line 401, de same à the same
Author Response
The authors would like to express our sincere gratitude to the reviewer for the time and effort devoted to evaluating our article. We believe that all the suggestions provided have significantly contributed to enhancing the clarity and quality of the manuscript. We have made every effort to address each of your comments thoroughly and accurately. We hope that our responses have satisfactorily resolved your concerns.
The research work entitled “Insertional Mutagenesis as a Strategy to Open New Paths in Microalgal Molybdenum and Nitrate Homeostasis” presents research on exploration of molybdenum uptake and nitrogen homeostasis pathways in green algae via insertional mutagenesis. Using this strategy, they identified 4 candidate genes from 5,200 mutants that may be involved in molybdenum transport, cofactor biosynthesis, and amino acid uptake. However, there are several concepts and diagrams proposed by the author in the article that needs to be properly modified, especially the graph in Fig.3 has repeated arrangement. Specific comments and general comments are given below:
Specific comments
- The logical flow of the Introduction appears somewhat disorganized. It is recommended to substantially revise this section by following a clearer structure: starting with the industrial significance of Chlamydomonas, followed by the importance of nitrogen uptake pathways in photosynthetic organisms, the role of molybdenum in these pathways, the current challenges in understanding molybdenum and nitrate homeostasis, and finally, the advantages of using the insertional mutagenesis strategy to address these gaps.
Thank you for your recommendation. We have reviewed the introduction as suggested. The major changes are marked in red in the text.
- Line 133-134, It is recommended to indicate the source of algae strains.
Thank you for your recommendation. Change to: The Chlamydomonas reinhardtii strains used in this study included the cell wall-deficient strain 704, which was obtained from the Chlamydomonas Resource Center (https://www.chlamycollection.org/).
- In Figure 3, the image in the second column for strain 704 (Ammonium-Par) appears to be identical to the image in the third column for strain 81.90 (normal-Mo). This is likely an unintentional oversight during figure preparation. The authors are requested to review the original data and replace the duplicated image with the correct one.
We are very grateful to the reviewer for their careful attention in checking Figure 3. We sincerely apologize for this mistake; we unintentionally copied and pasted the same image twice. The error has now been corrected
- Line 528-534, Since the composition of Tris-Acetate-Phosphate (TAP) medium is publicly available, it is recommended that the authors clearly state that even 100 nM of molybdenum is sufficient to support green algal growth. It would also be advisable to avoid describing this trace metal uptake characteristic using the term "contaminants," as it may lead to misunderstanding. A careful revision of this paragraph is suggested for improved clarity and accuracy.
We agree and have made the following change accordingly:
L577: This observation would imply that even when Mo is not intentionally added to the culture medium, its existing concentration is sufficient to sustain Chlamydomonas growth. This suggests that trace amounts of Mo may originate from impurities in other medium components, such as sulfate or phosphate salts, which are structural analogs of molybdate
- Line 561-581, The authors propose that Cre05.g234400 and Cre03.g191350 may function in molybdenum ion transport, citing the presence of specific characteristic domains. To further support this claim, it is suggested that the authors include sequence alignment results—perhaps as Figure 7—for visual reference. A similar recommendation applies to Cre08.g382545 and Cre08.g348040, as such data would strengthen the overall argument.
Thank you for your comment; we find it very appropriate and believe it improves the quality of the manuscript. To address it, we have created Figure 8 and referenced a phylogenetic study in which these transporters are classified into their subfamily.
L477: In Chlamydomonas, 75 ABC-type transporters have recently been identified and classified into the following subfamilies based on phylogenetic analysis: 7 ABCA, 8 ABCB, 10 ABCC, 3 ABCD, 1 ABCE, 11 ABCF, 26 ABCG, and 9 ABC [40]. In this study, the protein encoded by the gene Cre05.g234400 was classified as belonging to the ABCG subfamily, while the protein encoded by the gene Cre03.g1913500 was assigned to the ABCA subfamily. Figure 8A shows the analysis of conserved domains in the protein encoded by the gene Cre05.g234400. Two characteristic domains of the ABC family are present. First ABC domain: located between residues 558 and 758, with an E-value of 1.3e-22, indicating high statistical significance and suggesting strong conservation and functionality of this domain. Second ABC domain: located between residues 2203 and 2410, with an E-value of 3.9e-8, also statistically significant, although less conserved than the first. The presence of two separate ABC domains suggests that the protein may have a typical structure of full ABC transporters, which usually contain two ATP-binding domains. This supports the functional annotation of the gene as an ABC-type transporter. Figure 8B illustrates the analysis of conserved domains in the protein encoded by the Cre03.g191350 gene. Two well-defined ABC domains were identified: First ABC domain: located between residues 984 and 1169, with an E-value of 6.1e-18. This extremely low value indicates high statistical significance, suggesting strong functional conservation of this domain. Second ABC domain: located between residues 3323 and 3467, with an E-value of 2.9e-14, also highly significant and characteristic of ABC transporters. The considerable distance between both domains suggests the existence of intermediate regions, likely corresponding to transmembrane domains or spacer regions, which is common in this type of transporter.
- Li, X.; Li, X.; Yang, X.; Lan, C.; Huang, Y.; Jia, B. Identification and Characterization of ATP-Binding Cassette Transporters in Chlamydomonas Reinhardtii. Mar. Drugs 2022, 20, 63, doi:10.3390/md20100603.
And regarding the other two genes from our study:
L517: Figure 8C shows the conserved domain analysis of the protein encoded by the Cre08.g382545 gene. The most relevant finding is the identification of a ThiS domain located between residues 5 and 83, with an E-value of 4.1e-23, indicating very high statistical significance and strong evolutionary conservation of this domain. The ThiS domain is characteristic of proteins involved in Moco biosynthesis. The domain spans nearly the entire protein (residues 5 to 83 out of a total of 84), suggesting that the main function of the protein is directly associated with this domain.
L536: Figure 8D presents the conserved domain analysis of the protein encoded by the Cre07.g348040 gene, annotated as a BAT1-type amino acid permease. The most relevant finding is the identification of an AA permease 2 domain, located between residues 11 and 480, with an E-value of 1.2e-29, indicating very high statistical significance and strong evolutionary conservation of this domain. The AA permease 2 domain is characteristic of proteins that transport amino acids across the plasma membrane. The presence of this domain in the BAT1 protein suggests that it plays an essential role in amino acid uptake and exchange.
- Since the molybdenum and nitrate homeostasis pathway discussed in this article is a complex pathway, it is recommended that the author provide a schematic diagram to improve readability.
Thank you for your suggestion. We agree and have created a new figure, Figure 9, which we hope meets your expectations. This figure summarizes molybdenum metabolism, nitrate metabolism, and the possible roles of the proteins identified in this study.
General comments
- In the Materials and Methods section, the references to manufacturers are not formatted correctly. The proper citation format should be: (instrument model, manufacturer name, city, (state), country). For example, Line 160, (Microcellcounter F-500, Sysmex) à (Microcellcounter F-500, Sysmex, city, country). Other corrections required include line 180 (DNA Polymerase), line 196 (Gene Pulser), line 224 (NanoDrop), line 237 (restriction enzyme).
Thank you for your suggestion. We have adjusted the text accordingly:
(Countess 3 FL, Invitrogen, Waltham, USA)
DNA Polymerase (Thermo Scientific, Waltham, USA)
NanoDrop™ 2000/2000c spectrophotometer (Thermo Scientific, Waltham, USA)
Restriction enzyme AvaI (New England Biolabs, Ipswich, USA)
(Bio-Rad, Hercules, USA)
- Line 186 and 339, I would suggest revising the sentence to avoid using the future tense.
Thank you for your suggestion. We have adjusted the text accordingly:
L162: This DNA, hereafter referred to as the AphVIII marker, is used in the subsequent transformation experiment for insertional mutagenesis.
L316: This condition also helps us detect strains incapable of growing on nitrate. Additionally, two other media are used to further investigate Mo-related phenotypes.
- Line 207, 50 mM Tris-HCl, pH 8.0; 5 mM EDTA; 300 mM NaCl à 50 mM Tris-HCl, pH 8.0, 5 mM EDTA, 300 mM NaCl (semicolon à comma)
Thank you for your suggestion. We have adjusted the text accordingly:
The cell pellet was resuspended in 700 µL of lysis buffer (50 mM Tris-HCl, pH 8.0, 5 mM EDTA, 300 mM NaCl).
- Line 401, de same à the same
Thank you for your suggestion. We have adjusted the text accordingly:
under the same condition.

Reviewer 3 Report
Comments and Suggestions for Authors
Dear authors, thank you for the research and the manuscript. I have three minor comments that are more suggestions:
1) I would recommend exploring phylogenetic analyses of the two ABC-type proteins potentially involved in molybdate transport. The phylogenetic perspective could provide more context about these two identified proteins.
2) I would recommend considering the genomic context of the genes within the Chlamydomonas genome. This could provide valuable information about the genes surrounding those identified to determine if they have related functions, if they are conserved in the context of the genomes of other organisms, and/or information about their genomic dynamics, e.g. presence/absence in genomic alignments.
3) In the final part of the manuscript I would recommend including a sentence to complement your perspectives: perform transcriptome analyses under different experimental conditions, using varying concentrations of Mo.
Best regards
Author Response
The authors would like to express our sincere gratitude to the reviewer for the time and effort devoted to evaluating our article. We believe that all the suggestions provided have significantly contributed to enhancing the clarity and quality of the manuscript. We have made every effort to address each of your comments thoroughly and accurately. We hope that our responses have satisfactorily resolved your concerns.
Dear authors, thank you for the research and the manuscript. I have three minor comments that are more suggestions:
- I would recommend exploring phylogenetic analyses of the two ABC-type proteins potentially involved in molybdate transport. The phylogenetic perspective could provide more context about these two identified proteins.
Yes, we agree. However, the phylogenetic analysis of all the Chlamydomonas ABC transporter family members was conducted two years ago and published in Marine Drugs. According to the authors of that study, the two transporters identified in our work belong to the ABCA and ABCG subfamilies. To clarify this point, we have added the new figure 8 and the following text to the manuscript.
In Chlamydomonas, 75 ABC-type transporters have recently been identified and classified into the following subfamilies based on phylogenetic analysis: 7 ABCA, 8 ABCB, 10 ABCC, 3 ABCD, 1 ABCE, 11 ABCF, 26 ABCG, and 9 ABC [40]. In this study, the protein encoded by the gene Cre05.g234400 was classified as belonging to the ABCG subfamily, while the protein encoded by the gene Cre03.g1913500 was assigned to the ABCA subfamily. Figure 8A shows the analysis of conserved domains in the protein encoded by the gene Cre05.g234400. Two characteristic domains of the ABC family are present. First ABC domain: located between residues 558 and 758, with an E-value of 1.3e-22, indicating high statistical significance and suggesting strong conservation and functionality of this domain. Second ABC domain: located between residues 2203 and 2410, with an E-value of 3.9e-8, also statistically significant, although less conserved than the first. The presence of two separate ABC domains suggests that the protein may have a typical structure of full ABC transporters, which usually contain two ATP-binding domains. This supports the functional annotation of the gene as an ABC-type transporter. Figure 8B illustrates the analysis of conserved domains in the protein encoded by the Cre03.g191350 gene. Two well-defined ABC domains were identified: First ABC domain: located between residues 984 and 1169, with an E-value of 6.1e-18. This extremely low value indicates high statistical significance, suggesting strong functional conservation of this domain. Second ABC domain: located between residues 3323 and 3467, with an E-value of 2.9e-14, also highly significant and characteristic of ABC transporters. The considerable distance between both domains suggests the existence of intermediate regions, likely corresponding to transmembrane domains or spacer regions, which is common in this type of transporter.
- Li, X.; Li, X.; Yang, X.; Lan, C.; Huang, Y.; Jia, B. Identification and Characterization of ATP-Binding Cassette Transporters in Chlamydomonas Reinhardtii. Mar. Drugs 2022, 20, 63, doi:10.3390/md20100603.
2) I would recommend considering the genomic context of the genes within the Chlamydomonas genome. This could provide valuable information about the genes surrounding those identified to determine if they have related functions, if they are conserved in the context of the genomes of other organisms, and/or information about their genomic dynamics, e.g. presence/absence in genomic alignments.
Thank you for your comment. This is something we had already considered and analyzed previously, but it did not provide any supporting evidence. However, we agree that it is important to present and explain these data. Therefore, we have prepared the corresponding figure and have included an explanation in the Results section accordingly:
L462: We have analyzed the genomic context surrounding these two genes in the Chlamydomonas genome. As shown in Figure 7, the genes flanking each type of ABC-type transporter do not, when their functions are known, appear to be involved in the transport of any solute. Therefore, in this case, chromosomal localization does not provide evidence to support their function.
3) In the final part of the manuscript I would recommend including a sentence to complement your perspectives: perform transcriptome analyses under different experimental conditions, using varying concentrations of Mo.
We greatly appreciate your suggestion to perform transcriptome analyses under different experimental conditions using varying concentrations of Mo. In fact, this is an idea we had already considered, and we believe it represents a very promising direction. Therefore, we plan to pursue this as our next line of research. As suggested at the end of the manuscript, we have added these sentences
Regarding our future research on Mo homeostasis, we will focus on characterizing Mo transport activity in these mutants compared to the wild type. This approach will help elucidate the specific roles of these ABC transporters in Mo metabolism. Additionally, another strategy we plan to implement to identify new agents involved in Mo homeostasis is to perform transcriptome analyses under different experimental conditions, using varying concentrations of Mo.
L742: Regarding our future research on Mo homeostasis, we will focus on characterizing Mo transport activity in these mutants compared to the wild type. This approach will help elucidate the specific roles of these ABC transporters in Mo metabolism. Additionally, another strategy we plan to implement to identify new agents involved in Mo homeostasis is to perform transcriptome analyses under different experimental conditions, using varying concentrations of Mo.

Round 2
Reviewer 2 Report
Comments and Suggestions for Authors
I am satisfied with the revised manuscript.